# Tessellated 2D Convolution Networks: A Robust Defence against Adversarial Attacks

## Abstract

Data-driven (deep) learning approaches for image classification are prone to adversarial attacks. This means that an adversarial image which is sufficiently close (visually indistinguishable) from a true image of its representative class can often be misclassified to be a member of a different class. A reason why deep neural approaches exhibits such vulnerability towards adversarial threats is mainly because of the fact that the abstract representations learned in a data-driven manner often do not correlate well with human perceived features. To mitigate this problem, we propose the tessellated 2d convolution network, a novel divide-and-conquer based approach, which as a first step, independently learns the abstract representations of non-overlapping regions within an image, and then learns how to combine these representations to infer its class. It turns out that a non-uniform tessellation of an image which seeks to minimize the difference between the maximum and the minimum tessellated areas is the most robust way to construct such a tessellated 2d convolution network. This criterion can be achieved, among other schemes, by using a Mondrian tessellation of the input image. Experiments demonstrate that our proposed method of tessellated network provides a more robust defence mechanism against gradient-based adversarial attacks in comparison to conventional deep neural models.

## 1 Introduction

Deep neural networks are known to be susceptible to adversarial attacks. Image representations learned by a deep neural network differ from their visual interpretation. Attackers exploit this fact by introducing imperceptible evasive perturbation in a set of test images such that the victim network misclassifies them (Joseph et al., 2019). Defending neural networks against such adversarial attacks is of significant theoretical and practical importance. Major approaches to defence against such adversarial threats include adversarial training (Madry et al., 2018), network distillation (Papernot et al., 2016), input randomization (Xie et al., 2018), activation pruning (Dhillon et al., 2018), gradient masking (Goodfellow, 2018), input transformation (AprilPyone & Kiya, 2020), and ensemble methods (Tramèr et al., 2017) to name a few. Architectural changes in the network topology is a promising means of achieving adversarial robustness.

Well known evasive attacks include the gradient based input perturbation strategies such as fast gradient sign method (FGSM) (Goodfellow et al., 2015), and the projected gradient descent (PGD) (Madry et al., 2018) methodologies. Universal attacks that are image-agnostic and add the same perturbation for all input images while still modifying the class labels are also prevalent (Moosavi-Dezfooli et al., 2017). Norm based attacks seeking to optimize the perturbation were subsequently proposed to victimize newer defence strategies (Carlini & Wagner, 2017; Croce & Hein, 2019). Patch attacks, which involve perturbing image segments rather than the image pixels, have also been attempted Sharif et al. (2016). More recent attacking approaches include the use of ensembling-based strategies with a capability to adapt on the defence mechanisms employed (Tramèr et al., 2020).

Depending on the amount of information exposed to an attacker, an attack corresponds to one of the two types, namely i) black-box attack, those with little or no knowledge about the target model, and ii) white-box attack, where additional information about the network is available (e.g., in the form of architecture, optimization function used, model parameters etc.). A black-box attack often

involves substituting the victim network by a proxy network, constructed with the help of a small number of interactions with an oracle (Papernot et al., 2016). In between the two extremes of the black-box and white-box variants lies the gray-box attack, where the parameter values of a trained model are not available to an attacker; however, other information about the model (e.g., architecture details and optimization/activation functions) are available (Vivek et al., 2018). It has been reported that attacks methods can usually be effectively transferred to similar networks in a gray-box threat scenario. Numerous other threat scenarios like transfer-based, score-based, and decision-based black-box attacks are known in the literature (Ren et al., 2020; Dong et al., 2020).

As newer attacks are being developed, designing networks that are robust to adversarial attacks has been an ongoing game. Among the most popular defence mechanisms are the ones that are based on adversarial training using the samples generated by attacks such as FGSM Goodfellow et al. (2015) and PGD Madry et al. (2018) or their ensemble (Tramèr et al., 2017). State-of-the-art defences as reported in the RobustBench (Croce et al., 2020) benchmark dataset include those based on data augmentation for adversarial training (Rebuffi et al., 2021), as well as those that are based on transformation or randomization of model parameters (Gowal et al., 2021).

Various randomized image transformation schemes such as cropping, padding, compression, block segmentation, noise addition to convolution layer features demonstrate adversarial robustness (Xie et al., 2017; Liu et al., 2018; AprilPyone & Kiya, 2020) . Input rectification schemes attempts to remove adversarial perturbations by denoising, image blurring and depth reduction Xu et al. (2017). Transformation of the features at the output of the convolution layers like activation pruning (Goodfellow, 2018), denoising, are often equally effective Dhillon et al. (2018); Liao et al. (2018).

Regularization and dropout are recently being used for achieving adversarial robustness (Wang et al., 2018; B.S. & Babu, 2020; Jordão & Pedrini, 2021). A study on the effect of regularization and sparsity with respect to the adversarial robustness of a network can be found in (Schwartz et al., 2020; Pang et al., 2020). Generating diverse structured networks as a tool for robustness has been proposed in (Du et al., 2021; Pang et al., 2019). An alternative convolutional network (CNN) architecture which randomly masks parts of the feature maps also demonstrates adversarial robustness (Luo et al., 2020).

Architectural robustness provides an attractive defense mechanism that is *agnostic to attack strategies*. As a motivation of the work in this paper, we hypothesize that modification of the network structure leading to implicit feature transformation, cropping, masking, and distillation may result in improved robustness against adversarial attacks in an attacking method agnostic manner. Incorporation of diversity in network topology may also act as an effective defence against ensemble attacks. Consequently, reconfiguring the topology of a network may provide effective defence against adaptive adversarial attacks.

In this paper, we propose two dimensional *tessellated convolutional networks* (TCNN) that incorporates the effects of cropping, masking and feature transformation within a single framework. In our approach, an input image is partitioned into blocks (tiles) according to a tessellation (tiling) pattern. Each region of the input image makes use of a separate branch in the computation graph to propagate its effects forward in the form of feature representations. The individual feature representations then interact with each other for the eventual prediction of an image class (see Figure 1 for a schematic representation).

We investigate the use of three types of rectangular tessellation patterns, namely, regular grid tiling, tiling with non-uniform rectangles, and Mondrian partitioning (Roy et al., 2008) with a set of additional constraints on the rectangles. Existing research has applied Mondrian kernels for generating features Balog et al. (2016), and has also generalized Mondrian partitions for higher than 2 dimensions LeFevre et al. (2006).

Specifically, constraints in Mondrian tiling correspond to the following.

- The rectangular tiles are pairwise non-congruent, i.e., each rectangle must have a different dimension (widths and height values), e.g., a 2x8 rectangle can only be used only once. Note that this constraint does not prevent the use of another rectangle with a different dimension but identical, e.g., a 4x4 rectangle can be used in combination with a 2x8 one.
- The difference in the area of the largest and the smallest tiles is to be minimized. This difference is known as the score of the tiling.

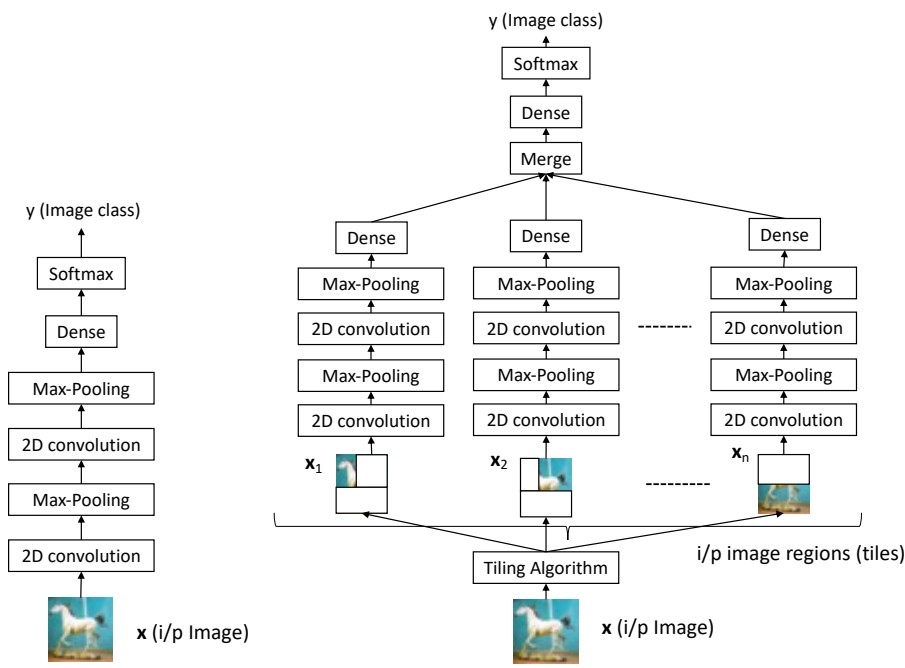

Figure 1: Architectural overview of the proposed approach of Tessellated 2D convolutional neural networks for defence against adversarial attacks on image classification networks. The left part of the figure depicts a standard layered 2D convolutional neural network for image classification. The right part of the figure shows our proposed method - 2D **Tessellated ConvNet (TCNN)**, where, first, the input $\mathbf{x} \in \mathbb{R}^{d \times d}$ is partitioned into $k$ sub-instances (non-overlapping image regions) - $\mathbf{x}_i$ such that $\cup_i \mathbf{x}_i = \mathbf{x}$.

While the first constraint ensures that there exist no parts in the overall computational graph with duplicate dimensions, the second one prevents solutions that employ too large or too small image regions. The implication of the former is that it is difficult for an adversarial attack to expose a vulnerable sub-network more than once (thus increasing the overall vulnerability), while the implication of the latter is that the individual feature representations of the tiles adequately represent meaningful parts of the overall image.

In our experiments, we find that 2D tessellated convolutional networks (2D-TCNNs) are more robust to well known attacks as compared to standard networks. Moreover, among the three different tiling approaches explored, a Mondrian tiling based 2D-TCNN leads to more robust results against adversarial attacks.

## 2 TESSELLATED CONVOLUTIONAL NETWORK

In this section, we describe our proposed method of Tessellated Convolutional Network (TCNN). We specifically focus on the 2D convolutional networks.

Figure 1 presents the idea of a TCNN. An input image is first partitioned into non-overlapping rectangular tiles using a tessellation scheme. Parallel branches of convolution and pooling layers of the tiled CNN then process each input segment. The convolution and pooling layers in each branch terminates in a dense layer of parameters leading to a feature representation of a part of the overall image corresponding to that tile. The output from these dense layers, each representing an abstract representation of a rectangular region of an image, are then concatenated in the merge layer and processed through yet another dense layer. The output layer is a softmax that is finally used to predict the discrete class label.

We use three tessellation schemes, namely, regular (or uniform), non-regular (or non-uniform), and Mondrian. Details of these partitions are presented next. The main focus of our paper is investigate

if the divide-and-conquer based approach of a tessellated network can lead to more robust defences against adversarial threats. For simplicity, we thus restrict our investigation to simple 2D convolution networks, instead of experimenting with more complex (in terms of depth and width) networks, e.g. ImageNet (Szegedy et al., 2015) or networks that use more involved connections between layers, e.g. the ResNet (He et al., 2016). However, our proposed divide-and-conquer based approach is generic enough to be applied on more complex computation graphs such as those of ImageNet or ResNet, which we leave as future work.

## 2.1 TESSELLATION METHODS

A tessellation of an $d \times d$ square image is a complete tiling of the image with non-overlapping tiles. Although the concept of tessellation can, in general, involve (even non-convex) polygons, the tiles, with which we cover an input image always refers to rectangles in the context of our problem.

A parameter to the tessellation process is the number of mutually disjoint rectangles used. Formally, each input $\mathbf{x} \in \mathbb{R}^{d \times d}$ is partitioned into $k$ sub-instances $\mathbf{x}_i$ such that $\cup_i \mathbf{x}_i = \mathbf{x}$.

To uniquely specify a tessellation of an input instance $\mathbf{x} \in \mathbb{R}^{d \times d}$, each tile $\mathbf{x}_i \in \mathbb{R}^{h_i \times w_i}$ of width $w_i$ and height $h_i$ is associated with a location, as specified by the row and column index of its top-left location, i.e., $p(\mathbf{x}_i) = (r_i, c_i)$ such that $1 \le r_i \le d - h_i$ and $1 \le c_i \le d - w_i$. Each tiling method, that we investigate, generates a list of such rectangular tiles.

### 2.1.1 REGULAR TESSELLATION

The simplest tessellation that we investigate is the uniform one, where each tile is a square. The parameter $k$ for regular rectangular tessellation controls the number of squares used to cover $\mathbf{x} \in \mathbb{R}^{d \times d}$, and is a perfect square, i.e., $k = m^2$ for some $m \in \mathbb{Z}$.

### 2.1.2 APERIODIC TESSELLATION

In non-uniform tessellation, an input image of size $d \times d$ is split into rectangular blocks of arbitrary sizes with a low likelihood that any two rectangles will be of equal area. We employ a recursive split and merge approach to generate a non-uniform tessellation. At each step we employ either a split or a merge operation depending on whether $m$ (the present number of tiles) is higher than or lower than $k$ (the desired number of tiles).

If $m < l$, we randomly select a rectangle and split it into two parts. The position of the splitting line and its direction (horizontal or vertical) is chosen randomly. The split operation always leads to increasing the total number of tiles by 1 (see Figure 2b for an illustrative example). Otherwise, if $m > l$, we merge a rectangle with other rectangles that are adjacent to it with respect to a direction (one of top, right, bottom or left). Figure 2c illustrates an example of merging a tile with the ones that are right-adjacent to it. The merge operation mostly leads to increasing the number of tiles.

We carry out a sequence of random split and merge operations on randomly selected tiles (sampled with uniform probability) unless the desired number of tiles (the parameter $k$) is reached. After every split or merge operation, we employ a post-hoc step which checks if any of the tiles is too small or too large (specifically, area less than $5^2$ or greater than $(3/4d)^2$). If the split or merge operation generates a rectangle whose area is either less than or higher than the two thresholds, then the step is undone.

It is also possible to generate non-uniform tessellations with other policies as well, e.g., with the use of Bayesian non-parametric space partition methods as surveyed in (Fan et al., 2021).

## 2.2 MONDRIAN TESSELLATION

The Mondrian tessellation is a non-uniform tessellation which is obtained by solving a constrained optimization problem. The objective is to minimize the difference between the area of the largest and the area of the smallest rectangle with the constraint that the rectangles are non-congruent to

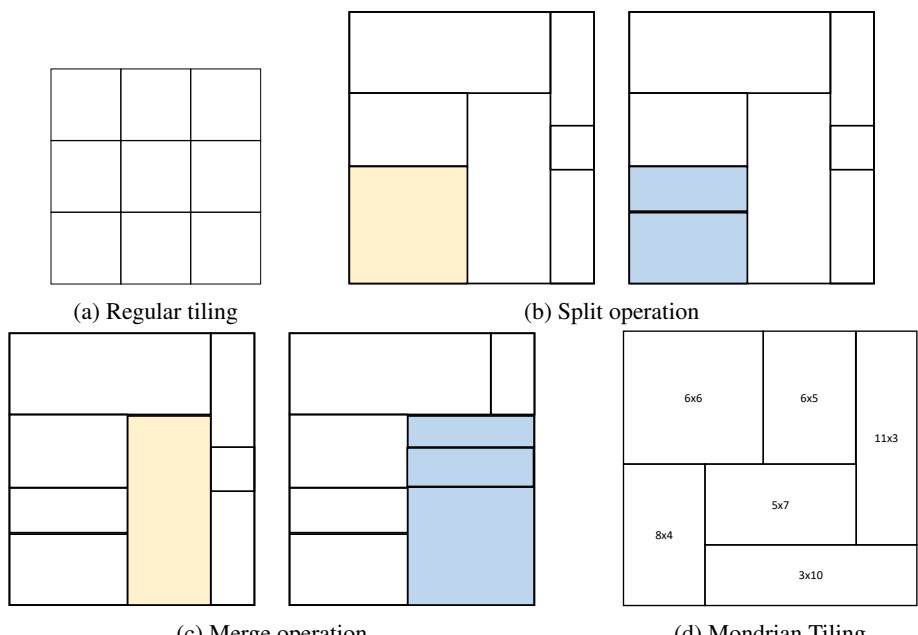

(a) Regular tiling

(b) Split operation

(c) Merge operation

(d) Mondrian Tiling

Figure 2: (a) Regular tiling of a $9 \times 9$ image with $9$ $3 \times 3$ square tiles ($d = 9$, $l = 9$); (b) Illustration of the effect of the *split* operation on a tiling - the yellow colored rectangle on the left part of the figure is transformed into two blue rectangles shown in the tiling shown along the right; (c) Illustration of the effect of the *merge* operation on a tiling - the yellow colored rectangle on the left part of the figure is merged with the rectangles right-adjacent to it; this operation for this example does not increase or decrease the total number of tiles (as seen on the right part of the figure); (d) A Mondrian tiling for a $14 \times 14$ square grid, note that the tiles are mutually non-congruent, the Mondrian score for this tiling is optimal, the value being $6 \times 6 - 3 \times 10 = 6$.

each other. Formally speaking, for a $d \times d$ square grid

$$\textbf{Minimize } s = \max_i w_i h_i - \min_j w_j h_j, \text{ s.t. } \nexists i, j : w_i h_i = w_j h_j \wedge (w_i - w_j)(w_i - h_j) = 0$$
$$1 \leq i \leq k, \ 1 \leq j \leq k, \ \forall i \ 1 \leq w_i \leq d, \wedge 1 \leq h_i \leq d. \tag{1}$$

where $w_i$ and $h_i$ denote the width and the height of the $i^{th}$ rectangle, and $k$ is the number of tiles in the current solution. The non-congruence constraint requires that if a rectangle with dimensions $a \times b$ has already been used in the tiling, then another rectangle of dimension $a \times b$ or $b \times a$ cannot be used.

The constrained optimization problem is solved with the following steps - first by enumerating all possible partitions of the number $d^2$ (the total area of the grid) by dynamic programming, second, by generating all possible factors of each element of the partition, and third, by generating the tiling by the Exact-Cover algorithm (Junttila & Kaski, 2010); see (Bassen, 2016) for more details.

For our problem, since images are of fixed size we did not require to find the optimal solution to the Mondrian problem for unknown values of grid sizes. Instead, as we work with the FMNIST and CIPHAR-10 images (of size 28x28 and 32x32, respectively), we used the known configuration of optimal Mondrian tiling as reported in (Bassen, 2016). In fact, we found that instead of using the solution for 28 and 32, splitting the original images into 4 blocks of 14x14 and 16x16 and employing four copies of optimal Mondrian tilings for 14x14 and 16x16 sizes yielded better results. Figure 2d shows the solution for the 14x14 block. Note that employing 4 copies of Mondrian solution for a $d/2 \times d/2$ grid eventually relaxes the global constraint of non-congruence of the tiles because in this case we allow duplicate tiles across the different $d/2 \times d/2$ blocks.

**Reason why tessellated network should work well against adversarial attacks.** The tessellated 2D convolution network dedicates a separate branch for each of the tiles as shown in Figure 1 which

are later combined in a dense layer. This offers an implicit ensemble of the features computed be each branch. Empirically, ensemble of diverse network structures have been shown to have robustness Du et al. (2021); Pang et al. (2019). Adopting a non-regular (and non-congruent) image tessellation scheme leads to diversity in the computational graph of the branches of tessellated CNN.

However, generation of non-regular tessellation patterns with large variation in sizes of the tiles may result in very large or too small sub-rectangles. Empirically we find that this harms the accuracy of the tessellated network on non-adversarial samples. Optimal and near optimal Mondrian patterns having low Mondrian scores (the value of $s$ in Equation 1) avoids such skew while using diverse shaped non-congruent sub-rectangles. This trade-off between performance on clean images and robustness to adversarial attacks has also been noted in previous studies. Both methods of generating non-uniform tessellations (Mondrian and those obtained with the split-merge operations) lead to better handling of this trade-off.

## 3 EXPERIMENTAL SETUP

In this section we present details of our experimental setting.

### 3.1 DATASETS

Two benchmark image classification datasets were used in our experiments experiments. Fashion-MNIST, variant of the standard MNIST dataset, which consists 10 classes each belonging to certain fashion category is used. Each image of the dataset is a $28 \times 28$ sized grayscale image. The training set consists of 60000 examples and a test set of 10000 examples. For a color image dataset, where adversarial perturbations are less perceptible, the CIFAR10 dataset was used. The CIFAR-10 dataset consists of 60000 $32 \times 32$ colour images in 10 classes, with 6000 images per class. There are 50000 training images and 10000 test images.

#### 3.1.1 ADVERSARIAL ATTACKS

Black-box untargeted versions of the two most standard adversarial attacks: FGSM (Fast Gradient Signed Method (Goodfellow et al., 2015)) and PGD (Projected Gradient Descent (Madry et al., 2018)) attacks were performed on the models for the purpose of the experiment.

For consistency, all the adversarial examples were generated apriori, and the same ones were used across every experiment. The FGSM examples were generated by applying $\ell_\infty$ perturbations of $\varepsilon = 8/255 = 0.03$, on normalized test sets. For the PGD examples as well, we used $\varepsilon = 0.03$, $num\_steps = 20$, and $step\_size = 2.5 \times \varepsilon / num\_steps$, where $num\_steps$ is the number of iterations of the PGD algorithm. In both the attacks we assume the victim network to be a standard 2D-CNN.

### 3.2 EVALUATION METRICS

To gauge the robustness of aforementioned models, it was necessary to compute and compare certain performance metrics, prior to and following the adversarial attacks.

Besides comparing classification accuracy (acc), a custom metric called weighted accuracy ($\mathrm{acc_w}$) was defined to represent the misprediction as well as the confidence of misprediction with a single numeric value. It is defined as :

$$\mathrm{acc_w} = 1 - \mathrm{mp} \cdot \mathrm{conf_{mp}},$$

where, $\mathrm{mp}$ denotes the mispredictions rate (*i.e.*, $1 - \mathrm{acc}$), and $\mathrm{conf_{mp}}$ denotes the average prediction confidence for misclassified examples. The confidence is measured in terms of normalized softmax output values. A low value of difference in the metrics for clean and adversarially perturbed input examples ($\Delta\mathrm{acc}$, $\Delta\mathrm{acc_w}$) denotes a better robustness.

### 3.3 TESSELLATION AND NETWORK PARAMETERS

In the proposed tessellated 2D convolution network, several experiments were performed over different tiling schemes, varying number of crops, different attacks, etc, to identify the most adversarially

resistant settings. For the fashion MNIST $28 \times 28$ images an optimal Mondrian tessellation for $n = 28$ was used with a total of $k = 24$ non-congruent sub-rectangles. The Mondrian score for this pattern is 9. The CIFAR10 $32 \times 32$ images were similarly split into four $16 \times 16$ blocks, for which known optimal Mondrian patterns are used consisting of a total $k = 20$ non-congruent rectangles. The Mondrian score for this solution is 9. We do not use Mondrian solutions for high values of $n$ to avoid large Mondrian score solutions even though they are optimal. In this process we sacrifice non-congruence in favor of lower Mondrian score. For regular tessellations we split the image into $(k = 4 \times 4 = 16)$ equal sized square blocks for all the experiments in both datasets. For aperiodic tessellation we split the input images into $k = 22$ sub-rectangles for fashion MNIST, and $k = 20$ sub-rectangles for CIFAR10 using the spiral pattern generation process described in Section 2.1.2. Note that, owing to the random initialization of the aperiodic spiral pattern generation process, multiple tessellations with identical values of $k$ may be generated. We report the average value of the evaluation parameters over all such configurations in Section 4.

The network layer structures for each of the CNN and the tessellated CNNs (Figure 1) considered in our experiments for the fashion MNIST dataset are mentioned below:

1. Basic CNN: 28 3x3 Conv; Pool 2x2; 56 3x3 Conv; Pool 2x2; 56 3x3 Conv; Flatten & Concat; Dense 512; Dense 128; Dense 10.

2. Regular tessellated CNN (with $4 \times 4 = 16$ sub-rectangles): Crop-layers (16 parallel); 28 3x3 Conv; Pool 2x2; 56 3x3 Conv; Pool 2x2; 56 3x3 Conv; Flatten & Concat; Dense 512; Dense 128; Dense 10.

3. Aperiodic tessellated CNN (with $22$ sub-rectangles): Crop-layers (all parallel custom crops); 28 3x3 Conv; Pool 2x2/2x1/1x2/1x1; 56 3x3 Conv; Pool 2x2/2x1/1x2/1x1; 56 3x3 Conv; Flatten & Concat; Dense 512; Dense 128; Dense 10.

4. Mondrian tessellated CNN (with $24$ sub-rectangles): Crop-layers (all parallel custom crops); 28 3x3 Conv; Pool 2x2/2x1/1x2/1x1; 56 3x3 Conv; Pool 2x2/2x1/1x2/1x1; 56 3x3 Conv; Flatten & Concat; Dense 512; Dense 128; Dense 10.

For the aperiodic and Mondrian tessellations, in the custom maxpooling layers with window sizes $2 \times 2$, $2 \times 1$, $1 \times 2$ or $1 \times 1$ options are used, (instead of the usual $2 \times 2$) to avoid pooling across a dimension of a tile, if that dimension size is too small.

The categorical cross-entropy is used as the loss function, and the Adam optimizer is used for training with a batch size of 32. All models were trained for 10 epochs.

## 4 RESULTS

Results from the experiments performed under different settings defence for $\ell_\infty$ perturbation of $\varepsilon = 8/255 = 0.03$ FGSM (Goodfellow et al., 2015) and PGD (Madry et al., 2018) black-box attacks is reported in Table 1. The goal of these experiments is to study if a partitioning method applied on a specific network structure offers a significant improvement over the "non-partitioned" counterpart. It was observed that the drops in classification accuracy as well as in weighted-accuracy, both in the cases of FGSM and PGD attacks, happens to significantly reduce for the tessellated convolutional network as compared to vanilla 2D Convolution Network. We further observed that the aperiodic and Mondrian methods seem to have lower accuracy drop than regular tessellation in most cases.

The number of sub-rectangles $k$ used in a tessellation is a hyperparameter of our model. We show a plot of the variation in classification accuracy for clean and FGSM attacked test samples for the fashion MNIST dataset in Figure 3(a) for regular, aperiodic, and Mondrian tessellations. The drop in accuracy is also plotted in Figure 3(b). A higher $\ell_\infty$ perturbation of $\varepsilon = 0.04$ was considered to study the effect of hyperparameter variation. We study the effect of variation in hyperparameter $k$ only for the regular and aperiodic tessellations; since, the number of rectangles are fixed in Mondrian patterns. The accuracy and drop in accuracy values for the Mondrian tessellations ($k = 24$) are shown as a baseline in the figure. It is seen that the drop of accuracy has a valley when around 22 number of sub-rectangles are used in aperiodic tiling, and 16 sub-rectangles for regular tiling. The robust accuracy for attacked samples is less when a very low number of rectangles are used. On the other hand the drop in accuracy increases when a very large number of sub-rectangles are used due to increase in skewness. Similar trends were observed for other datasets and attacks.

Table 1: Performance of Tessellated 2D convolutional neural networks against $\ell_\infty$, $\varepsilon = 8/255$ FGSM and PGD adversarial attacks. Hyperparameter values of the number of sub-rectangles in the tessellation for the FMNIST dataset are: regular ($k = 16$), aperiodic ($k = 22$), Mondrian ($k = 24$). Values for the CIFAR10 dataset are: regular ($k = 16$), aperiodic ($k = 20$), Mondrian ($k = 20$). Bold typeset values denote the best performance across networks for each metric and each attack type. Bold underlined values shows the best $\Delta$acc and $\Delta$acc$_w$ metrics for the PGD attack for each dataset.

| Network | Tessellation | Attack | FMNIST | | | | CIFAR-10 | | | |
| --- | --- | --- | --- | --- | --- | --- | --- | --- | --- | --- |
| | | | acc | $\Delta$acc | acc$_w$ | $\Delta$acc$_w$ | acc | $\Delta$acc | acc$_w$ | $\Delta$acc$_w$ |
| 2D-CNN | N/A | None | **0.9136** | ____ | **0.9360** | ____ | **0.7337** | ____ | **0.8281** | ____ |
| 2D-CNN | N/A | FGSM | 0.7864 | 0.1272 | 0.8350 | 0.1010 | 0.3989 | 0.3348 | 0.5794 | 0.2487 |
| 2D-CNN | N/A | PGD | 0.7552 | 0.1584 | 0.8057 | 0.1303 | 0.3291 | 0.4046 | 0.5241 | 0.3040 |
| 2D-TCNN | Regular | None | 0.9080 | ____ | 0.9279 | ____ | 0.6527 | ____ | 0.7810 | ____ |
| 2D-TCNN | Regular | FGSM | **0.8617** | **0.0463** | 0.8908 | 0.0371 | 0.5119 | 0.1408 | 0.6834 | 0.0976 |
| 2D-TCNN | Regular | PGD | **0.8549** | **0.0531** | **0.8851** | 0.0428 | 0.5006 | 0.1521 | 0.6694 | 0.1116 |
| 2D-TCNN | Aperiodic | None | 0.9104 | ____ | 0.9320 | ____ | 0.6487 | ____ | 0.7868 | ____ |
| 2D-TCNN | Aperiodic | FGSM | 0.8585 | 0.0519 | **0.8909** | 0.0411 | **0.5166** | **0.1321** | **0.6951** | **0.0917** |
| 2D-TCNN | Aperiodic | PGD | 0.8503 | 0.0601 | 0.8841 | 0.0480 | 0.5011 | 0.1476 | 0.6788 | 0.1080 |
| 2D-TCNN | Mondrian | None | 0.9029 | ____ | 0.9242 | ____ | 0.6409 | ____ | 0.7797 | ____ |
| 2D-TCNN | Mondrian | FGSM | 0.8543 | 0.0486 | 0.8834 | 0.0408 | 0.5080 | 0.1329 | 0.6880 | **0.0917** |
| 2D-TCNN | Mondrian | PGD | 0.8469 | 0.0560 | 0.8768 | **0.0474** | **0.5017** | **0.1392** | 0.6789 | **0.1008** |

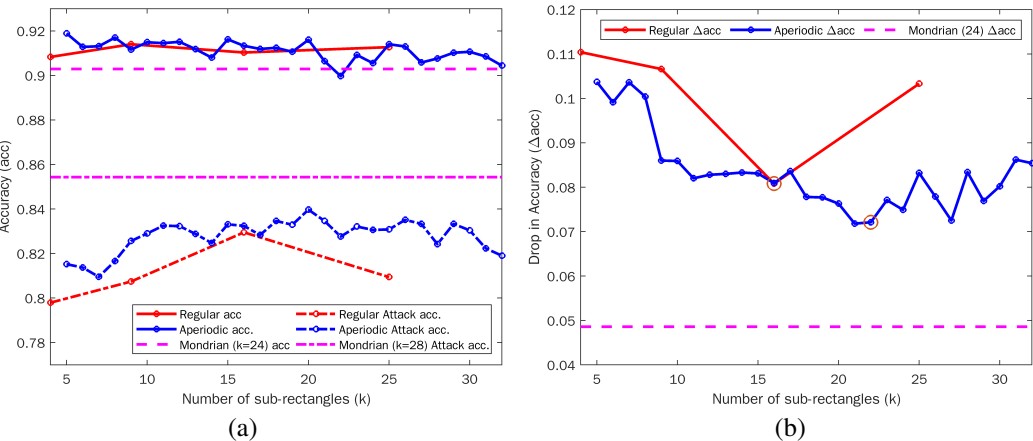

(a)             (b)

Figure 3: Variation of (a) Classification accuracy (acc) on clean, and FGSM attacked ($\ell_\infty$, $\varepsilon = 0.04$) test examples, and (b) drop in accuracy ($\Delta$acc), with the number of sub-rectangles (hyperparameter $k$) in for the fashion MNIST dataset. The circled points in (b) correspond to smallest $\Delta$acc for regular and aperiodic tessellations. Since, $k = 24$ is fixed in Mondrian tessellation we show it as a baseline in the plots.

We compare the performance of the proposed tessellated convolution network (T-CNN) with some of the well known defence strategies for the gradient based black-box untargeted attacks FGSM and PGD. We report results for the CIFAR10 dataset. The following methods were compared:

- 2D-CNN
- Regular tessellated 2D-CNN (T-2D-CNN)
- Aperiodic tessellated 2D-CNN (T-2D-CNN)
- Mondrian tessellated 2D-CNN (T-2D-CNN)
- Adversarial trained (A-T) ResNet using FGSM samples (Goodfellow et al., 2015)
- Adversarial trained (A-T) ResNet using PGD samples (Madry et al., 2018)

Table 2: Accuracy (acc), and drop in accuracy ($\Delta$acc) of compared models for FGSM and PGD adversarial attacks for CIFAR10 dataset. Bold typeset values indicate best $\Delta$acc for an attack.

| Network | None (acc) | FGSM (acc) | FGSM ($\Delta$acc) | PGD (acc) | PGD ($\Delta$acc) |
|---|---|---|---|---|---|
| 2D-CNN | 0.7337 | 0.3989 | 0.3348 | 0.3291 | 0.4046 |
| Regular ($k = 16$) T-2D-CNN | 0.6527 | 0.5119 | 0.1408 | 0.5006 | 0.1521 |
| Aperiodic ($k = 20$) T-2D-CNN | 0.6487 | 0.5166 | 0.1321 | 0.5011 | 0.1476 |
| Mondrian ($k = 20$) T-2D-CNN | 0.6409 | 0.5080 | 0.1329 | 0.5017 | **0.1392** |
| A-T FGSM ResNet | 0.8740 | 0.9090 | **-0.0350** | 0.2105 | 0.6635 |
| A-T PGD ResNet | 0.7940 | 0.5170 | 0.2770 | 0.4370 | 0.3570 |
| PixelDefend ResNet | 0.7900 | 0.3985 | 0.3915 | 0.2989 | 0.4911 |
| I-T ResNet | 0.7214 | 0.6665 | 0.0549 | 0.4030 | 0.3184 |

- PixelDefend generative model based training data augmentation for ResNet (Song et al., 2018), and
- Input transformed (I-T) (cropping, quilting, total variance maximization) test examples and ResNet (Guo et al., 2018).

The classification accuracy are shown in Table 2. Even though the clean and robust accuracy of the ResNet based models are higher compared to CNN based ones due to their higher capacity, we observe lesser drop in accuracy after attack ($\Delta$acc) in the proposed tessellated CNN models.

## 5 CONCLUSIONS AND FUTURE WORK

We present tessellated 2D convolutional network as a divide and conquer defence against adversarial attacks for image classification. The network first partitions the input image into a number of non-overlapping sub-rectangles. Each sub-rectangle is then processed by a parallel branch of the CNN terminating in dense layers. The output of these branches are concatenated and passed through another dense layer to obtain the final softmax classification scores. Three input image partitioning strategies namely, regular, aperiodic, and Mondrian is considered. We achieve a good degree of robustness in this approach as compared to non-tessellated networks. The implicit parallelization of the computation graphs into diverse branches while maintaining balanced sized partitions produces feature transformations that leads to high classification accuracy on clean data while providing adversarial robustness.

Studies on other space partitioning techniques may help in devising more diverse classes of tessellated networks. Similarly, tessellation structures for other networks like residual nets and wide residual nets may be considered in future.

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
