# OpenReview forum: "Tessellated 2D Convolution Networks: A Robust Defence against Adversarial Attacks"
_ICLR.cc/2022/Conference — ICLR 2022 Submitted_

### Official Review · Reviewer_CSai · 2021-10-16

**Correctness:** 4
**Technical Novelty And Significance:** 3
**Empirical Novelty And Significance:** 3
**Recommendation:** 5
**Confidence:** 5

**Main Review:**

Strengths:
- The paper is clearly written.
- The proposed method is quite simple, straightforward, and looks to be effective. I wish the authors explain more about the rationale behind this partitioning idea. My personal hypothesis is: the smaller the patch, the less "space" the attacker has to mess up the prediction. However, the smaller the patch, the vanilla network also has less context to generate a accurate prediction. So there should be a sweet spot. This paper basically shows that the sweet spot is not the original image size: rather it's k ~ 20.

Weaknesses:
- My #1 concern is that the method is not validated on larger datasets such as ImageNet. The stories for 32x32 images may or may not hold for 224x224 images. I encourage the authors to compare against works such as [1].
- My #2 concern is that the method is not validated on larger capacity networks, such as ResNet variants listed in Table 2. For example, A-T FGSM ResNet significantly outperforms all tessellated variants, whether it's clean accuracy or accuracy after FGSM attack. A 2D CNN that achieves 0.7337 accuracy on CIFAR-10 is not really competitive. Public implementations go well above 0.9, and the state-of-the-art is easily above 0.95.
- There are a couple of places throughout the paper with "FMNIST" and "CIPHAR-10". Are these typos of "MNIST" and "CIFAR-10"?


[1] Xie, Cihang, et al. "Feature denoising for improving adversarial robustness." Proceedings of the IEEE/CVF Conference on Computer Vision and Pattern Recognition. 2019.

**Summary Of The Paper:**

This paper studies defense against adversarial attacks. In particular, the proposal is to partition the image into non-overlapping sub-rectangles, run a separate CNN on each, and finally merge to generate the prediction. The defense effectiveness is evaluated on MNIST and CIFAR-10.

**Summary Of The Review:**

I really enjoy the idea of the paper. It provides evidence that the hypothesis I gave in the Main Review above may be true, which can inspire more research in adversarial defense. However, though the current experimental results on MNIST and CIFAR-10 are encouraging, I have two major concerns regarding the data and the network capacity, respectively. On the bright side, if the authors can show that the story holds for larger images and stronger networks, then the paper is much stronger and more generally impactful.

---

### Official Review · Reviewer_Znf4 · 2021-11-02

**Correctness:** 2
**Technical Novelty And Significance:** 3
**Empirical Novelty And Significance:** 1
**Recommendation:** 3
**Confidence:** 4

**Main Review:**

### Strengths
It is interesting that this paper proposes to make the computation of each subnetwork heterogeneous by splitting the input into tiles. The reviewer believes that this approach may be novel in this regard.

### Weaknesses
* This paper did not provide sufficient justification for the choice of tiling methods.
	* The paper did not provide insights into why different tiling schemes could bring different robustness impacts.
	* Specifically, it is perceivable that tiling with different random seeds with the same number of tiles would incur a high degree of variance in the evaluated accuracies. It is important to repeat the experiments and report the mean and variance of the results. This is especially problematic for the Mondrian tiling as it appears that the results only used one such possible tiling for Figure 3a and b, and for Table 2.
* The choices of the adversarial attacks are also problematic as recent strong attacks are excluded from the empirical results. Please consider using [1-4] for more accurate evaluation of robustness. FGSM is simply too weak, and PGD is slow to converge and both could provide a false sense of robustness, especially only 20 iterations were used for each image.
* In Section 2.2 the authors provided “reasons why tessellated network should work well against adversarial attacks”. It mentions that “ensemble of diverse network structures have been shown to have robustness”, citing [5]. Unfortunately, it was recently shown in [6] that the method proposed in [5] is not robust under attacks (accuracy can reach 0% under $\ell_\infty$-bounded with perturbation $\epsilon = 0.01$).

### Minor issues
* It is not clear if the models are obtained after adversarial training or not. The reviewer was unable to find explanation for this in the paper.

### References
```
[1]: Croce et al., Reliable evaluation of adversarial robustness with an ensemble of diverse parameter-free attacks. ICML 2020. http://proceedings.mlr.press/v119/croce20b.html
[2]: Yu et al., LAFEAT: Piercing Through Adversarial Defenses with Latent Features. CVPR 2021. https://openaccess.thecvf.com/content/CVPR2021/papers/Yu_LAFEAT_Piercing_Through_Adversarial_Defenses_With_Latent_Features_CVPR_2021_paper.pdf
[3]: Tashiro et al., Diversity Can Be Transferred: Output Diversification for White- and Black-box Attacks. NeurIPS 2020. https://arxiv.org/abs/2003.06878
[4]: Brendel et al., Accurate, reliable and fast robustness evaluation. NeurIPS 2019. https://proceedings.neurips.cc/paper/2019/file/885fe656777008c335ac96072a45be15-Paper.pdf
[5]: Pang et al., Improving Adversarial Robustness via Promoting Ensemble Diversity. ICML 2019. https://arxiv.org/abs/1901.08846
[6]: Tramer et al., On Adaptive Attacks to Adversarial Example Defenses, NeurIPS 2020. https://proceedings.neurips.cc/paper/2020/file/11f38f8ecd71867b42433548d1078e38-Paper.pdf
```

**Summary Of The Paper:**

This paper proposes to split the input image into tiles that are fed into separate subnetworks for feature extraction, and to form an implicit ensemble by combining the results from the subnetworks for a final prediction. The paper claims that such ensemble can improve adversarial robustness of the model.


**Summary Of The Review:**

While the idea of tessellation for heterogenous subnetworks is interesting, the paper is lacking in thorough empirical evidence to back up the claims of robustness.

---

### Official Review · Reviewer_m2PC · 2021-11-03

**Correctness:** 2
**Technical Novelty And Significance:** 2
**Empirical Novelty And Significance:** 2
**Recommendation:** 3
**Confidence:** 4

**Details Of Ethics Concerns:**

No novel ethical issues are raised by this paper.

**Main Review:**

The paper presents a novel architecture for image classification using image tessellation. This architecture is interesting, and warrants more investigation, but the central claim of the paper, namely that this architecture results in better adversarial robustness, is not sufficiently supported by the experimental results.

Crucially, the authors do not show white box results, and do not specify on which architecture the black box attacks were derived. If the black box attacks were generated from a standard neural net, as opposed to a tessellated one, then the central claim of "improved adversarial robustness" should be reduced to "reduced transferability of standard conv net adversarial attacks". It is well established in the literature that changes to the network architecture result in less black box transferability, and so although this architecture is novel, this result is not.  If it is instead the case that the authors derived the black box attacks from similarly tessellated neural networks than the central claim should be one of black box transferability within network architectures. This is interesting but warrants more work to understand why tesselation reduces black box transferability. Either Way the claim of a "robust defence mechanism against gradient-based adversarial attacks" is unfounded.


**Summary Of The Paper:**

This paper presents a new architecture for image classification neural networks based on various forms of image tessellation. They split the input image up into numerous patches, feed each patch through the 2d convolutional network and then combine the representations of each patch before a final dense classification layer.  They show that this architecture results in better 'black box' adversarial robustness.

**Summary Of The Review:**

This work presents an interesting architecture, but does not support the central claim of adversarial robustness because they do not present white box results, and do not specify on which architecture the black box attacks were derived.

---

### Official Review · Reviewer_HrGf · 2021-11-03

**Correctness:** 3
**Technical Novelty And Significance:** 3
**Empirical Novelty And Significance:** 2
**Recommendation:** 5
**Confidence:** 3

**Main Review:**

The idea of this paper is simple and intuitive. The paper is well organized and easy to follow.

The result seems quite effective as shown in the experiment. The intuition of why it works makes sense to me.

However, there are two major concerns of the method that makes me uncertain of this method.

The first concern is whether this method is still effective for a high-resolution image. Whether the performance gain is still significant. For a higher resolution image, there are usually more margins outside the object; the object is not as tightly cropped, and the object size (proportional to the image) varies a lot. This is quite different from fashion MNIST and CIFAR10. In this scenario, the requirement of tiles number might need to increase or harder to find a good strategy for tiling if the object size varies too much. The overhead will also be an issue if the number of tiles needs to be significantly increased in that situation.

The second concern is the tiling will lose important spatial information, which might make tasks that depend on the spatial relationship suffer. For example, human action classification will require the spatial relationship between body parts.

I am also curious about whether this method is effective for the more localized attacks, like patch-attack. In that situation, the attacked region might happen within the same tile.

**Summary Of The Paper:**

This paper presents a tessellated convolution network that is more robust to adversarial attacks. In the paper, authors proposed three different ways of tiling. Experiments show this method is more robust to FGSM and PGD for classification tasks on fashion MNIST and CIFAR10.


**Summary Of The Review:**

The idea is interesting, simple, and intuitive. Experiments show that it is effective for the classification task.
However, the two major concerns make me uncertain about the effectiveness of this method.
I believe this idea has good potential. More experiments and discussions can make the argument much stronger. If these two concerns can be properly addressed, this will be quite impactful work.

---

### Decision · Program_Chairs · 2022-01-20

**Decision:**

Reject

**Comment:**

This paper proposes an image tesselation scheme to improve the robustness of image classifiers.  The reviewers agree that the method is simple and intuitive, and view this as a positive attribute.  At the same time, the reviewers want to see if the method works on higher resolution images.  It was also not clear to reviewers how the attacks on the method were constructed, whether they were white box, and whether they were adaptive.  Without a rebuttal, these questions remain unanswered.